# Conjugated dual size effect of core-shell particles synergizes bimetallic catalysis

Xiaohui Zhang[1,3], Zhihu Sun [2,3], Rui Jin[1,3], Chuwei Zhu[1,3], Chuanlin Zhao[1], Yue Lin [1], Qiaoqiao Guan[1], Lina Cao [1], Hengwei Wang [1], Shang Li[1], Hancheng Yu[1], Xinyu Liu[1], Leilei Wang[1], Shiqiang Wei [2] ✉, Wei-Xue Li [1] ✉ & Junling Lu [1] ✉

Core-shell bimetallic nanocatalysts have attracted long-standing attention in heterogeneous catalysis. Tailoring both the core size and shell thickness to the dedicated geometrical and electronic properties for high catalytic reactivity is important but challenging. Here, taking Au@Pd core-shell catalysts as an example, we disclose by theory that a large size of Au core with a two monolayer of Pd shell is vital to eliminate undesired lattice contractions and ligand destabilizations for optimum benzyl alcohol adsorption. A set of Au@Pd/SiO$_2$ catalysts with various core sizes and shell thicknesses are precisely fabricated. In the benzyl alcohol oxidation reaction, we find that the activity increases monotonically with the core size but varies nonmontonically with the shell thickness, where a record-high activity is achieved on a Au@Pd catalyst with a large core size of 6.8 nm and a shell thickness of ~2−3 monolayers. These findings highlight the conjugated dual particle size effect in bimetallic catalysis.

Bimetallic catalysts have drawn tremendous attention in widespread applications, including fine chemical synthesis, biomass conversion, and fuel cells, for their large phase spaces in various compositions and structures to optimize their activity and selectivity[1–4]. In particular, mixing a precious metal with an abundant one in the form of an alloy or core-shell structure improves the cost efficiency necessary for commercialization. Compared to alloy catalysts, core-shell catalysts have peculiar lattice strains and ligand effects to optimize the geometric and electronic properties[2,3,5]. The ligand effects and charge transfers from the core are usually localized at the core-shell interface and are therefore sensitive to the shell thickness. The thin metal shell itself might also exhibit a prominent quantum size effect (QSE). Meanwhile, lattice strains in the metal shell often occur owing to its lattice mismatch with the core at the interface and usually persist over several atomic layers[5,6]. These three factors could be entangled with each other, making

important impacts on the overall reactivity. This is particularly true for lattice strains, which often show significant modulations of the shell electronic properties and corresponding reactivity[1–3,5,7–9].

Great efforts have been devoted to tuning lattice strains to boost catalytic performance[1–3,5–7,9,10]. Previous studies have mainly focused on the variation of the composition and structure of the metal core to control the lattice strains[5,6,9,10]. However, the particle size could also play essential roles in bimetallic catalysis. In particular, as the core size is reduced, the metal core often possesses considerable lattice contractions and changes in electronic properties (e.g., work functions)[11–15]. These changes can in fact significantly interfere with the lattice strains and charge transfers in the shell, thus causing remarkable impacts on the bimetallic synergy. Addressing the lattice strain by varying the core particle size to optimize the shell reactivity is inspiring but has not yet been explored to the best of our knowledge.

[1]Department of Chemical Physics, Hefei National Research Center for Physical Sciences at the Microscale, Key Laboratory of Surface and Interface Chemistry and Energy Catalysis of Anhui Higher Education Institutes, Collaborative Innovation Center of Chemistry for Energy Materials (iChEM), University of Science and Technology of China, Hefei 230026, China. [2]National Synchrotron Radiation Laboratory, University of Science and Technology of China, Hefei 230029, China. [3]These authors contributed equally: Xiaohui Zhang, Zhihu Sun, Rui Jin, Chuwei Zhu. ✉e-mail: sqwei@ustc.edu.cn; wxli70@ustc.edu.cn; junling@ustc.edu.cn

Consequently, investigation of the core size on tailoring the lattice strains and the influence of the shell thickness on the ligand effects and QSE is highly valuable for further improving the overall reactivity of the core-shell catalysts and fundamentally understanding these individual roles.

Here, we address the above critical questions using Au@Pd core-shell catalysts for their superior performance in the reaction of benzyl alcohol (BzOH) oxidation to benzyl aldehyde, an important industrial process for fine chemical synthesis[16]. To disentangle the strain and ligand effects modulated by the core size and shell thickness, we first studied the adsorption of BzOH on various Au@Pd catalysts by density functional theory (DFT) calculations. It is found that the adsorption on Au@Pd increases with Au size due to the alleviation of lattice contraction in the core. The ligand effects from the core destabilize the adsorption on the Pd shell but become negligible when the Pd thickness increases from 1 to 2 monolayers (MLs) with increasing adsorption. For a further increase in the shell thickness, however, the adsorption is weakened due to the downshift of the surface Pd $\varepsilon_{4d}$ centers. Guided by theory, we combined the deposition–precipitation (DP) method with atomic layer deposition (ALD) to fabricate a large set of Au@Pd core-shell NPs on a SiO$_2$ support by precisely regulating both the Au core size from 2.8 to 6.8 nm and the Pd shell thickness from approximately 0.3 to 3.2 MLs. In the reaction of solvent-free BzOH oxidation, the trend of the overall activity on the Au size and Pd thickness coincides perfectly with the calculated trend of the BzOH adsorption, and a record high activity of $6.86 \times 10^4$ h$^{-1}$ is achieved for the 6.8 nm Au core and ~2–3 ML Pd shell. The present work provides critical insight to disentangle the involved geometric and electronic properties of core-shell catalysts and highlights the importance of fabricating a dedicated structure and composition to boost the activity and selectivity.

## Results and discussion
### Theoretical insights

DFT calculations were performed to understand the strain and ligand effects in Au@Pd core-shell catalysts modulated by the Au size and Pd thickness. Note that compared to the lattice constant of bulk Pd in a face-centered cubic (fcc) structure, the calculated and experimental values of bulk Au (fcc) are 5.5% and 4.9% larger, respectively, and significant stretching of the Pd shell over the Au core is expected. We constructed different Au clusters from Au$_{55}$ (~1.3 nm), Au$_{147}$ (~1.7 nm), Au$_{309}$ (~2.2 nm), Au$_{561}$ (~2.7 nm) to Au$_{923}$ (~3.2 nm) in a cuboctahedron shape, where isotropic contractions with respect to bulk are found (Supplementary Table 1) and plotted in Fig. 1a. The average contraction in the (111) in-plane lattice constant and interlayer spacing is significant at 4.97% and 5.26% for small Au$_{55}$, respectively, which almost offsets the difference in the lattice constants between bulk Au and Pd. The contraction decreases with increasing cluster size and remains considerable at 3.32 and 3.50% for Au$_{923}$. Within the Au clusters constructed, the calculated values are consistent with the electron microscopic observation of small Au particles (Fig. 1b and Supplementary Fig. 1 and Supplementary Table 2) and previous results[11-15]. Although a further increase in the cluster size is beyond the available computational resources, our experiments show that the contraction decreases continuously with increasing Au particle size and becomes negligible on approximately 6.8 nm Au particles, which will be shown later.

To model Au@Pd$_{1ML}$ core-shell catalysts with different sizes, the outmost atomic layer in Au$_{55}$, Au$_{147}$, Au$_{309}$, Au$_{561}$, and Au$_{923}$ was replaced with Pd, which was noted as Au$_{13}$@Pd$_{42}$, Au$_{55}$@Pd$_{92}$, Au$_{147}$@Pd$_{162}$, and Au$_{309}$@Pd$_{252}$ (Fig. 1c and Supplementary Figs. 2, 3), respectively. The optimized Pd–Pd distance (in average) is 2.72 Å for the small Au$_{13}$@Pd$_{42}$ cluster and increases gradually to 2.78 Å for the large Au$_{309}$@Pd$_{252}$ cluster, which is close to that in pristine Pd(111) (2.79 Å). To probe the influence on reactivity, the adsorption of BzOH

was considered, and the favorable adsorption site for Au$_{13}$@Pd$_{42}$ was identified at the low-coordination corner site with a calculated adsorption energy of −2.59 eV. With an increase in the cluster size, the same adsorption site was adopted to observe the pure size-induced strain effect (Supplementary Fig. 4). For the larger cluster of Au$_{55}$@Pd$_{92}$, there is a considerable enhancement in adsorption energy (−2.88 eV). Whereas for further increase of the size to Au$_{147}$@Pd$_{162}$ and Au$_{309}$@Pd$_{252}$, the enhancement becomes small with adsorption energies of −2.96 and −2.90 eV (Fig. 1d), respectively. The increased adsorption with the increase in cluster size could be rationalized by the reduction in lattice contraction, a fact that is well documented in the literature[3,6,10]. In fact, for the same reason, the BzOH adsorption on bulk truncated clusters with larger lattice constants is stronger than that on the relaxed clusters (Supplementary Fig. 5). To model BzOH adsorption on large Au$_{NP}$@Pd$_{1ML}$ (6.8 nm at least), where the lattice contraction in the Au core is completely diminished, we constructed a pristine four-layer Au(111) surface with a 1 ML Pd overlayer, denoted Au(111)@Pd$_{1ML}$. Although the Pd–Pd lattice in the corresponding Pd overlayer is approximately 5% larger than that in Au$_{309}$@Pd$_{252}$, the calculated adsorption energy is −2.87 eV, which is 0.03 eV weaker. The slight weakening is because the gain in adsorption energy from lattice stretching is counteracted by the higher coordination number on the flat (111) terrace than on Au$_{309}$@Pd$_{252}$.

In addition to the lattice strain, the ligand effects also play an important role in BzOH adsorption. We found that for Au(111)@Pd$_{1ML}$, the subsurface metal right below the topmost Pd layer is Au, which actually destabilizes the overall adsorption toward BzOH. To see this clearly, we removed the Au substrate underneath from Au(111)@Pd$_{1ML}$ and calculated the corresponding adsorption energy of BzOH without structural optimization (the blue curve in Fig. 1d). Therein, the BzOH adsorption on Pd$_{1ML}$ without the Au substrate becomes much stronger (−3.80 eV) and otherwise weakens to −2.87 eV. In line with Au(111)@Pd$_{1ML}$, the role of the Au core in Au$_{NP}$@Pd$_{1ML}$ also destabilizes the BzOH adsorption on Pd$_{1ML}$. With a decrease in the cluster size, the extent of destabilization induced by the Au substrate becomes smaller for small clusters of Au$_{13}$@Pd$_{42}$, Au$_{55}$@Pd$_{92}$, and Au$_{147}$@Pd$_{162}$. We note that although the corresponding charge redistribution increases slightly (Fig. 1e and Supplementary Fig. 6) and the work function decreases (Supplementary Fig. 7), there are no obvious changes in the calculated Bader charge for adsorbed BzOH. Accordingly, the variation in destabilization of BzOH adsorption on the small Au$_{NP}$@Pd$_{1ML}$ clusters is attributed to the different lattice contractions (Fig. 1a, b).

The ligand effects from the Au substrate are, however, localized at the Au–Pd interface only, and destabilization diminishes when the Pd thickness increases from 1 ML to 2 MLs. The subsurface metal right below the topmost Pd overlayer is Pd instead of Au, and the adsorption energy becomes the strongest of −3.34 eV for Au(111)@Pd$_{2ML}$. In fact, for Au(111)@Pd$_{nML}$ ($n \geq 2$), the presence of the Au substrate has little influence on the calculated adsorption energy (Fig. 1d). Nevertheless, the BzOH adsorption shows a pronouced QSE on the Pd thickness. It is weakened gradually with a further increase in the Pd thickness, and there is a very small change in adsorption energy (approximately −2.86 eV) for $n \geq 4$. To rationalize the result, we note that with an increase in the Pd thickness from 2 MLs to 6 MLs, the surface Pd $\varepsilon_{4d}$ shifts downward from −1.64 to −1.74 eV with respect to the Fermi level, and similarly, the subsurface Pd $\varepsilon_{4d}$ shifts downward from −2.01 to −2.25 eV (Supplementary Fig. 8 and Supplementary Table 3). These downshifts lower the corresponding reactivity and weaken the molecule adsorption. For comparison, on the four-layer pristine Pd(111) surface, whose in-plane lattice constant is 5.4% smaller than that on Au(111)@Pd$_{4ML}$, the surface Pd atom has a deep-lying $\varepsilon_{4d}$ of −1.96 eV, and BzOH has an even weaker adsorption energy of −2.70 eV. The BzOH adsorption on pristine Au(111) is the weakest at −1.21 eV (Supplementary Fig. 9).

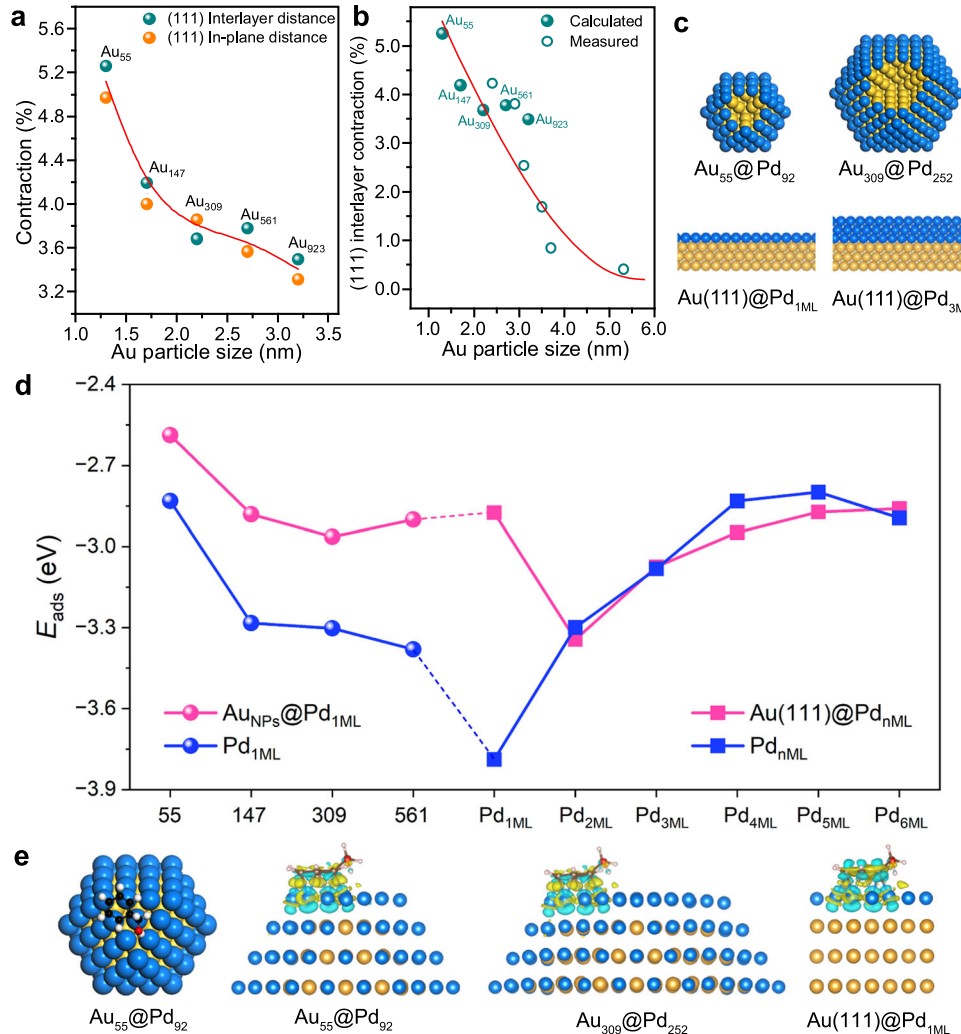

**Fig. 1 | Theoretical insight into BzOH adsorption on Pd-based catalysts.**
**a** Calculated contractions for (111) interlayer spacing and in-plane distance (in average) for various Au clusters relative to that of Au bulk. **b** Comparison of calculated contractions for (111) interlayer spacing (on average) over various Au clusters/particles with those measured by transmission electron microscopy. **c** Models of representative $Au_{NPs}@Pd_{1ML}$ clusters and $Au(111)@Pd_{nML}$ slab models; the yellow and blue balls are Au and Pd atoms, respectively. **d** The adsorption energies of BzOH on $Au_{NP}@Pd_{1ML}$ (pink circle) and $Au(111)@Pd_{nML}$ (pink square), and the blue counterparts represent the results without the presence of the Au core inside (for $Au_{NPs}@Pd_{1ML}$ clusters) or the Au substrate underneath (for $Au(111)@Pd_{nML}$ slabs). **e** Top views of BzOH adsorption on $Au_{55}@Pd_{92}$ (blue = Pd, gold = Au, black = C, white = H, red = O) and the charge density difference contours upon BzOH adsorption on $Au_{55}@Pd_{92}$, $Au_{309}@Pd_{252}$, and $Au(111)@Pd_{1ML}$. The navy blue and yellow contour areas represent the depletion and accumulation of electrons, respectively.

## Synthesis and structure characterization of Au@Pd core-shell catalysts

To examine the dual size effect, a large set of Au@Pd core-shell bimetallic catalysts with different Au core sizes and Pd shell thicknesses were precisely synthesized by combining the DP method with selective Pd ALD (Fig. 2a). First, four Au/SiO$_2$ catalysts with Au particle sizes of 2.8 ± 0.5, 4.3 ± 0.6, 5.6 ± 1.2, and 6.8 ± 0.9 nm were synthesized using the DP method by varying the temperature and pH values (denoted as x nm-Au)[17], as verified by transmission electron microscopy (TEM) and X-ray diffraction (XRD) (Supplementary Figs. 10, 11). After that, Pd ALD was executed on the Au catalysts at 150 °C to deposit Pd exclusively on the Au NPs. Negligible Pd deposition on the SiO$_2$ support under the same ALD conditions was attributed to the innerness of SiO$_2$ to the nucleation of the Pd precursor[18,19], as confirmed by inductively coupled plasma atomic emission spectroscopy (ICP–AES) (Supplementary Fig. 12 and Supplementary Table 4). Gradual tailoring of the Pd shell thickness was realized by varying the number of Pd ALD cycles, and the as-prepared samples are denoted as Au$_x$@yML-Pd, where x and y represent the size of the Au core in nanometers and the number of Pd shell layers calculated according to the cuboctahedron

cluster model, respectively (Supplementary Table 5)[20]. Aberration-corrected high-angle annular dark-field scanning TEM (HAADF-STEM) measurements of Au$_x$@2.9ML-Pd catalysts showed that the average particle size after Pd deposition increased by approximately 1.3–1.6 nm to 4.1 ± 0.6, 5.7 ± 0.9, 7.2 ± 1.1, and 9.4 ± 1.6 nm for Au$_{2.8}$@2.9ML-Pd, Au$_{4.3}$@2.9ML-Pd, Au$_{5.6}$@2.9ML-Pd, and Au$_{6.8}$@2.9ML-Pd, respectively (Fig. 2b–e). Given that the thickness of one ML Pd is approximately 0.224 nm[21], the thicknesses of the Pd shells in these Au$_x$@2.9ML-Pd catalysts were ~2.8–3.5 MLs, in good agreement with those calculated according to the cluster model (Supplementary Figs. 13, 14, Supplementary Tables 4, 5). The Pd deposition rate on Au NPs was approximately 0.05–0.06 nm per cycle (or 0.22–0.26 ML per cycle) in line with the literature[18,19]. Energy-dispersive spectroscopy (EDS) elemental mappings showed that the Au signals were surrounded by Pd signals (Fig. 2f–h). XRD showed that there was no shift in the locations of the diffraction peaks regardless of the Pd shell thickness (Supplementary Fig. 15). UV–visible (UV–Vis) spectra also revealed a gradual attenuation of the characteristic surface plasmon resonance peak of Au particles at 517 nm as the Pd coverage increased (Supplementary Fig. 16)[22]. The above three techniques all

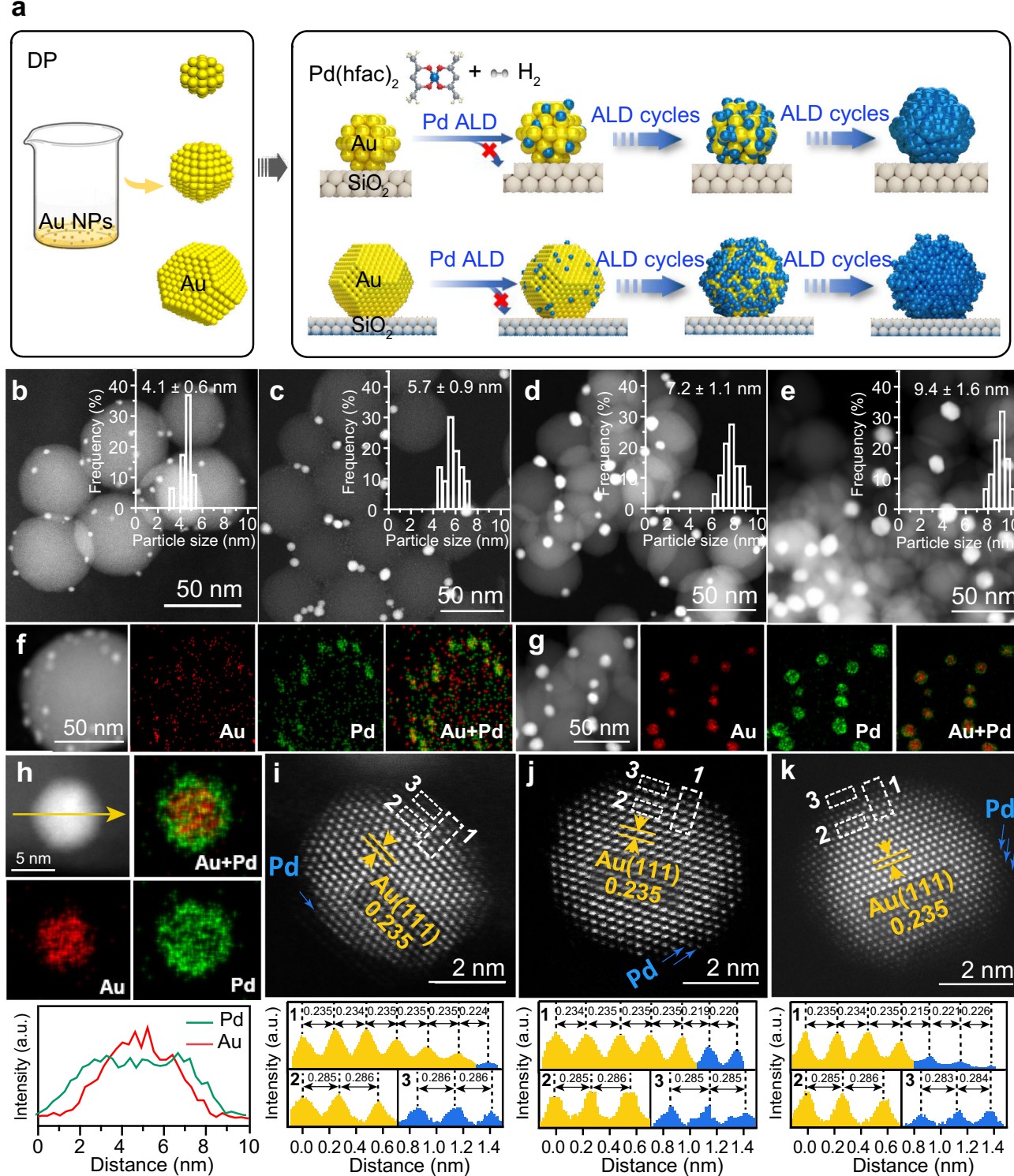

**Fig. 2 | Synthetic illustration and morphology of Au@Pd core-shell catalysts.**
**a** Schematic illustration of the synthesis of Au$_x$@$y$ML-Pd core-shell bimetallic cat-
alysts with varied Au core sizes and Pd shell thicknesses by combining the DP
method with selective Pd ALD. HAADF-STEM images of Au$_{2.8}$@2.9ML-Pd (**b**),
Au$_{4.3}$@2.9ML-Pd (**c**), Au$_{5.6}$@2.9ML-Pd (**d**), and Au$_{6.8}$@2.9ML-Pd (**e**) at low magni-
fications. The insets show their corresponding particle size distributions. STEM
image and the corresponding EDS elemental mapping of Pd Lα1, Au Lα1, and
constructed Pd + Au signals of Au$_{2.8}$@2.9ML-Pd (**f**) and Au$_{6.8}$@2.9ML-Pd (**g**) at low
magnifications as well as a Au@Pd particle in Au$_{6.8}$@2.9ML-Pd (**h**) at high magni-
fication. Line profiles across the Au@Pd particle in (**h**) along the yellow arrow are
also demonstrated. Representative atomic-resolution HAADF-STEM images of
Au$_{6.8}$@1.1ML-Pd (**i**), Au$_{6.8}$@2ML-Pd (**j**), and Au$_{6.8}$@2.9ML-Pd (**k**), along with the
corresponding line intensity profiles along the numbered dashed rectangles to
show the distance of the interplanar distance and lattice distance.

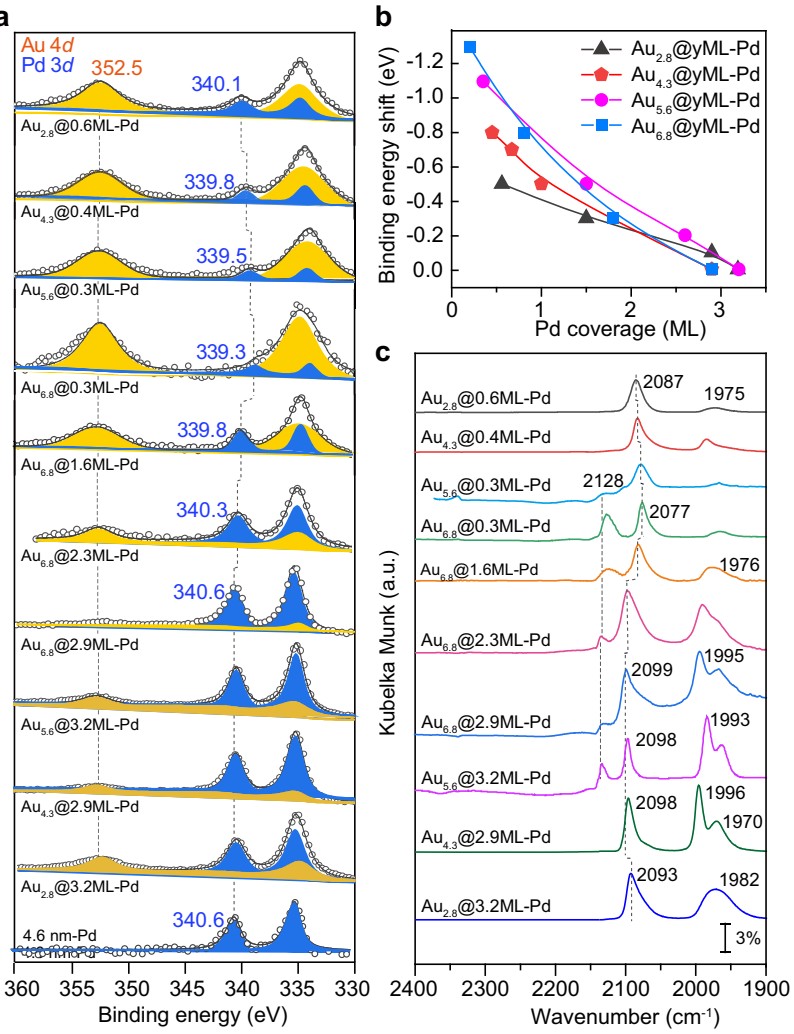

**Fig. 3 | Electronic properties of Au@Pd core-shell catalysts. a** XPS spectra of $Au_x$@$y$ML-Pd bimetallic catalysts and 4.6 nm-Pd in the Pd $3d$ and Au $4d$ regions. **b** Plots of the shifts of Pd $3d_{5/2}$ binding energy as a function of Pd coverage in the four sets of $Au_x$@$y$ML-Pd catalysts normalized to that of bulk Pd. **c** DRIFT spectra of CO chemisorption on $Au_x$@$y$ML-Pd catalysts at saturation coverage.

unambiguously confirmed the formation of the Au@Pd core-shell structure.

To obtain deep insight into the core-shell structure, a representative atomic-resolution HAADF-STEM image of $Au_{6.8}$@1.1ML-Pd is shown in Fig. 2i. The presence of an ~1 ML Pd shell was confirmed by the relatively darker brightness due to its smaller Z value than Au and the decrease in the interplanar distance between the outermost and the second layers to 0.224 nm by 4.7% compared to those (0.235 nm) in the interatomic layers and those in monometallic 6.8 nm Au NPs (Supplementary Fig. 17)[23,24]. The in-plane lattice distance of the epitaxial Pd layer was ~2.86 Å close to the underlying Au lattice, manifesting a lattice expansion of 4% compared to bulk Pd (2.74 Å), which is in excellent agreement with theoretical calculations. Similarly, a decrease in the interplanar distance was observed in the outer two and three layers of $Au_{6.8}$@2ML-Pd and $Au_{6.8}$@2.9ML-Pd particles, respectively (Fig. 2j, k), suggesting that the Pd shell thicknesses were approximately 2 and 3 MLs in these two samples, respectively. The Pd dispersions in these bimetallic catalysts were measured using CO as a probe molecule. We found that the Pd dispersions by CO titration were consistent with the cuboctahedron cluster model in the set of $Au_{2.8}$@$y$ML-Pd catalysts but were relatively smaller in the other three sets of $Au_x$@$y$ML-Pd catalysts with larger Au core sizes, especially at higher Pd coverages (Supplementary Fig. 18 and Supplementary Table 4). This result suggests that for the larger Au cores, interdiffusion

between Au and Pd might have occurred to endow the formation of less sharp Au–Pd interfaces by forming a Au-rich core and Pd-rich shell structure, in line with the literature[25,26].

X-ray photoemission spectroscopy (XPS) measurements were employed to investigate the electronic properties of these catalysts. As shown in Fig. 3a, the 4.6 nm Pd sample showed binding energies of Pd $3d_{5/2}$ and $3d_{3/2}$ located at 335.3 and 340.6 eV, respectively, indicating zero-valence Pd[27]. For Au@Pd bimetallic catalysts, the two peaks of Pd $3d$ and Au $4d$ partially overlap in this region. Deconvolution of the XPS spectrum of $Au_{2.8}$@0.6ML-Pd showed that the Pd $3d$ doublets of $3d_{5/2}$ and $3d_{3/2}$ were located at 334.8 and 340.1 eV, respectively, and the Au $4d$ doublets of $4d_{5/2}$ and $4d_{3/2}$ were located at 334.8 and 352.5 eV, respectively[28]. Focusing on the isolated Pd $3d_{3/2}$ peak, $Au_{2.8}$@0.6ML-Pd exhibited a considerable downward shift of Pd $3d$ binding energy by 0.5 eV with respect to that of 4.6 nm-Pd, consistent with the suggested electronic interactions in the AuPd bimetallic system, where Au usually gains $s$, $p$ electrons and loses $d$ electrons while Pd loses $s$ and $p$ electrons but gains $d$ electrons[29,30]. Increasing the Au core size at a similar submonolayer Pd coverage (denoted as $Au_x$@SubML-Pd), the downward shift became substantially greater up to 1.3 eV observed on $Au_{6.8}$@0.3ML-Pd, manifesting the remarkable enhancement of electronic interactions between Au and Pd by increasing the Au core size. In contrast, we found that the Au $4d_{3/2}$ binding energy was almost unchanged at 352.5 eV for all these samples. On the other hand,

increasing the Pd coverage, the Pd $3d_{3/2}$ peak gradually shifted back from 339.3 to 339.8, 340.3, and 340.6 eV on the set of $Au_{6.8}$@0.3ML-Pd, $Au_{6.8}$@1.6ML-Pd, $Au_{6.8}$@2.3ML-Pd, and $Au_{6.8}$@2.9ML-Pd catalysts, respectively (Fig. 3a). A similar trend of Pd binding energy shift on Pd coverage was observed for all sets of Au@Pd bimetallic catalysts, in line with the literature (Supplementary Fig. 19)[19]. In contrast, the Au $4f$ binding energy peaks gradually shifted to lower values with increasing Pd coverage (Supplementary Fig. 20). Meanwhile, we also noticed that the intensity of the Au $4d$ and $4f$ peaks gradually decreased with increasing Pd coverage (Fig. 3a and Supplementary Figs. 19, 20), indicating the formation of a Au@Pd core-shell structure, which agrees well with the STEM, XRD, and UV–Vis results (Fig. 2, Supplementary Figs. 15, 16). At a high Pd coverage of ~3 MLs, the Pd $3d$ binding energy for all Au@Pd catalysts measured was nearly identical to each other, regardless of the Au core size. To clearly see the shifts of Pd $3d_{5/2}$ binding energy in these four sets of bimetallic catalysts, we further plotted the Pd $3d_{5/2}$ binding energy shifts relative to zero-valence Pd (335.3 eV) as a function of Pd coverage (Fig. 3b). It is clear that the larger Au core and the thinner Pd shell rendered a more negative binding energy shift, while the shift turned to level off at the high Pd coverage of ~3 MLs, regardless of the Au core size. It is obvious that the Au core size and Pd shell thickness together shape the electronic properties of the Pd shell in these Au@Pd core-shell bimetallic catalysts.

The electronic and geometric properties of Pd shells in Au@Pd bimetallic catalysts were further investigated by diffuse reflectance infrared Fourier transform spectroscopy (DRIFTS) with CO as a probe molecule. As shown in Fig. 3c, $Au_{2.8}$@0.6ML-Pd exhibited a dominant peak at 2087 cm$^{-1}$ and a much weaker peak at 1975 cm$^{-1}$, assigned to linear and bridge-bonded CO on Pd, respectively[31]. The weak bridge-bonded CO feature indicates that the Pd atoms were atomically dispersed and surrounded by Au atoms in the majority in this sample[19,32]. As the Au core size increased to 6.8 nm but at a similar low Pd coverage, the linear CO peak on Pd exhibited a continuous redshift, reaching 2077 cm$^{-1}$ on $Au_{6.8}$@0.3ML-Pd; meanwhile, a new peak at 2128 cm$^{-1}$ appeared, assigned to linear CO on low-coordination Au atoms[33,34]. The larger redshift of the linear CO peak on $Au_{6.8}$@0.3ML-Pd than on $Au_{2.8}$@0.3ML-Pd suggests that the Pd atoms in $Au_{6.8}$@0.3ML-Pd have more populated $4d$ state electrons, which strengthens the Pd($4d$)-CO($2\pi^*$) bonding through $\pi$-backdonation, thus endowing a redshift of the CO peak, in excellent agreement with the XPS results in Fig. 3a. With an increase in the Pd coverage for all four sets of Au@Pd bimetallic catalysts, the bridge-bonded CO peak (below 2000 cm$^{-1}$) developed aggressively and became the dominant peak at high Pd coverages (Fig. 3c and Supplementary Fig. 21), clearly demonstrating the gradual evolution of Pd on Au NP surfaces from isolated Pd atoms or very small aggregates to large ensembles and to continuous islands or continuous Pd shells, in line with HAADF-STEM results in Fig. 2. In addition, we also noticed that the feature of linear CO on low-coordination Au atoms was present in $Au_{6.8}$@2.9ML-Pd and $Au_{5.6}$@3.3ML-Pd but absent in $Au_{4.3}$@2.9ML-Pd and $Au_{2.8}$@2.9ML-Pd, implying interdiffusion between Au and Pd by forming a Au-rich core and Pd-rich shell structure, consistent with the CO titration experiments (Supplementary Fig. 18 and Supplementary Table 4).

In situ X-ray absorption fine structure (XAFS) spectroscopy measurements were further performed at both Pd $K$ and Au $L_3$ edges to reveal the coordination environments of Pd and Au atoms in the Au@Pd catalysts. At the Pd $K$-edge, the X-ray absorption near edge structure (XANES) curves of all these Au@Pd catalysts exhibited a shape similar to that of Pd foil (Supplementary Fig. 22), but the absorption edge shifted slightly to lower energies. Therein, the catalysts with a larger Au core showed a larger shift to lower energies, suggesting the electron-richness of Pd atoms in the Au@Pd bimetallic catalysts, in line with the XPS and CO DRIFTS observations (Fig. 3). As the Pd coverage on 6.8 nm Au

increased, the downward shift gradually returned and became close to that of Pd foil, again in line with the XPS results.

Fourier transforms of the Pd $K$-edge extended X-ray absorption fine structure (EXAFS) spectra in the real ($R$) space demonstrated that at the submonolayer coverage of Pd, the curves of $Au_x$@SubML-Pd showed a doublet peak at -2.24 and 2.85 Å (Fig. 4a). This feature is distinctly different from that of Pd foil, which exhibits a prominent Pd–Pd peak at 2.5 Å with a side lobe at 2.0 Å due to the nonlinear phase shift of Pd scattering, indicating the presence of Pd–Au bonds in $Au_x$@SubML-Pd in the majority. With an increase in the Au core size, the doublet peak shifted slightly to larger $R$ values, implying a longer Pd-Au bond distance for the Au@Pd bimetallic samples with a large Au core. The Fourier transformed EXAFS curve fittings revealed that in all the Au@Pd samples, the coordination numbers (CNs) of Pd-Au and Pd-Pd are ~8.0 and ~1.0, respectively (Fig. 4b, Supplementary Figs. 23, 24 and Supplementary Table 6), suggesting that the Pd atoms in $Au_x$@SubML-Pd are within the outermost surface Au layer with near atomic dispersion (inset of Fig. 4b). Increasing the Pd coverage in $Au_{6.8}$@$y$ML-Pd increased the intensity of the peak at 2.85 Å, while the doublet peak shifted to lower $R$ values (Fig. 4a). EXAFS curve fittings showed a gradual decrease in Pd-Au CNs to 3.5, 3.3, and 2.5 and an increase in Pd-Pd CNs to 6.0, 7.9, and 9.1 for $Au_{6.8}$@1.1ML-Pd, $Au_{6.8}$@2ML-Pd, and $Au_{6.8}$@2.9ML-Pd, respectively. Meanwhile, the total Pd CNs slightly increased from 9.0 on $Au_{6.8}$@0.3ML-Pd to 11.6 on $Au_{6.8}$@2.9ML-Pd, suggesting the evolution of Pd species on Au from atomic dispersion to large ensembles and to a continuous Pd shell (inset of Fig. 4b), consistent excellently with the STEM and CO DRIFTS observations (Figs. 2, 3c). The length of Pd-Pd coordination in $Au_{6.8}$@2.9ML-Pd was elongated to 2.77 Å compared to that of Pd foil (2.74 Å), in line with the lattice expansion in the Pd shell observed by STEM and theoretical calculations (Figs. 1b, 2i–k). Here, the Pd-Au CNs in $Au_{6.8}$@1.1ML-Pd, $Au_{6.8}$@2ML-Pd, and $Au_{6.8}$@2.9ML-Pd were slightly larger than the corresponding values of 3.0, 1.4, and 0.9 estimated according to the cubic-octahedral cluster model (Supplementary Fig. 24). This result implies that interdiffusion of Au and Pd to a certain extent occurred to cause an obscure Au–Pd interface, thus increasing the Pd–Au coordination (inset of Fig. 4b), in line with the CO titration and DRIFTS CO chemisorption measurements (Fig. 3d, Supplementary Figs. 18, 21). A similar conclusion was also reached for $Au_{5.6}$@3.2ML-Pd and $Au_{4.3}$@2.9ML-Pd.

At the Au $L_3$-edge, the XANES spectra of the Au@Pd bimetallic samples showed a much smaller difference from each other, and all were close to the curve of Au foil, suggesting that Au was mostly in the metallic state (Supplementary Fig. 22), in line with the XPS results (Fig. 3a). The Fourier transformed EXAFS spectra in the $R$ space showed that all the Au@Pd bimetallic samples exhibited a triplet peak in the region of 1.8–3.3 Å (Fig. 4c). The triplet peaks of the $Au_x$@SubML-Pd samples had a shape similar to that of Au foil. While increasing the Pd coverage on the 6.8 nm Au core, the shape of the triplet peak changed considerably by increasing the intensity of the peak at 2.15 Å but decreasing the intensity of the peak at 2.64 Å, which suggests a strong deconstructive interference of Au–Au scattering by Au-Pd scattering[35]. EXAFS curve fittings show that the CNs of Au–Au coordination were all above 10, while the Au–Pd CNs were smaller than 1.9 even at high Pd coverages (Fig. 4d, Supplementary Fig. 25 and Supplementary Table 7), further verifying the Au@Pd core-shell structure. Here, the Au–Pd CNs in the samples of $Au_{6.8}$@$y$ML-Pd with high Pd coverages are also considerably higher than the values of ~0.7 estimated according to the cubic-octahedral cluster model (Supplementary Fig. 24), again confirming the Au-rich core and Pd-rich shell structure. $Au_{5.6}$@3.2ML-Pd and $Au_{4.3}$@2.9ML-Pd showed a similar result.

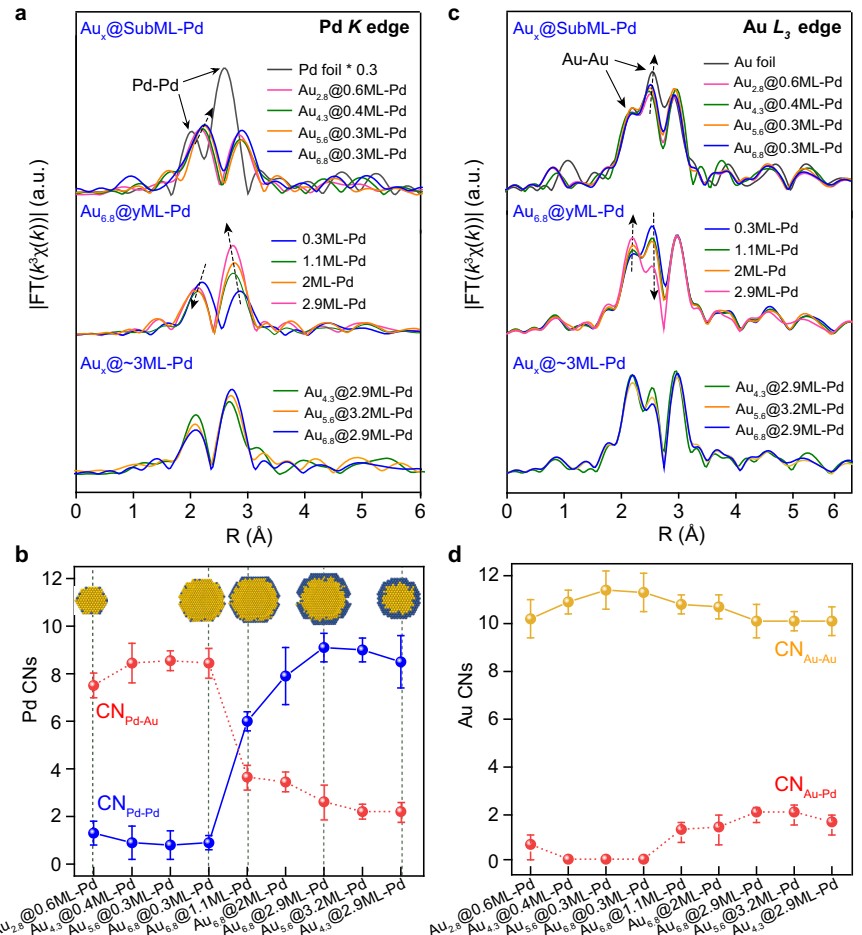

**Fig. 4 | Coordination structures of Au@Pd bimetallic catalysts.** Detailed microstructural evolution information shown in the Fourier transformed $k^3$-weighted $\chi(k)$ of the Pd $K$-edge (**a**) and Au $L_3$-edge (**c**) of the Au$_x$@$y$ML-Pd catalysts along with the reference metal foils. The Fourier transforms were not corrected for phase shifts. **b**, **d** Fourier transformed EXAFS fitting results of the metal-metal coordination numbers (CNs) for the Au$_x$@$y$ML-Pd bimetallic nanoparticles. Error bars represent the standard deviation. $R$ represents the distance between the absorbing atom and neighboring scattered atoms, and $\chi(k)$ denotes the amplitude of the EXAFS oscillations as a function of photoelectron wavenumber $k$. The inset in **b** and **d** shows the models of different sizes of Au$_{2.8}$@0.6ML-Pd, Au$_{6.8}$@0.3ML-Pd, Au$_{6.8}$@2.9ML-Pd, and Au$_{4.3}$@2.9ML-Pd bimetallic core-shell catalysts, respectively, where the dark blue and yellow balls are Pd and Au atoms.

## Catalytic performance evaluation

Selective oxidation of alcohols to the corresponding carbonyl compounds is an important industrial process for fine chemical synthesis[16]. Here, solvent-free selective oxidation of BzOH was utilized as a probe reaction to evaluate the dual size effect of these Au@Pd core-shell catalysts by conducting in a three-necked flask at 90 °C using $O_2$ as an oxidant. We found that all Au/SiO$_2$ catalysts were nearly inactive under this mild reaction condition (Supplementary Fig. 26), in line with the literature[36]. A Pd/SiO$_2$ catalyst with a Pd particle size of $4.6 \pm 0.6$ nm (denoted as 4.6 nm-Pd, Supplementary Fig. 27), an optimized Pd particle size for the oxidation of BzOH according to the literature[37,38], was also synthesized and evaluated. This 4.6 nm-Pd catalyst showed a turnover frequency (TOF) of approximately $0.32 \times 10^4$ h$^{-1}$, based on the number of Pd surface atoms, along with a benzaldehyde selectivity of ~90% (Supplementary Fig. 26). The much higher activity of 4.6 nm-Pd than those Au catalysts is in line with the much stronger adsorption of BzOH on Pd(111) ($-2.70$ eV) than on Au(111) ($-1.21$ eV). For Au@Pd core-shell bimetallic catalysts, we found that their intrinsic activities as well as specific activities based on the total Pd contents tightly depend on both the Au core size and Pd shell thickness (Fig. 5a and Supplementary Fig. 28), while the benzaldehyde selectivity was generally above 80% in all these Au@Pd bimetallic catalysts (Fig. 5b and Supplementary Fig. 29)[21]. By increasing the size of the Au core but maintaining the same Pd coverage, the activity increased substantially, regardless of Pd

coverage. This agrees excellently with the same trend of the increase in BzOH adsorption from $-2.47$ eV on Au$_{137}$@Pd$_{1ML}$ to $-2.87$ eV on Au(111)@Pd$_{1ML}$ (Fig. 1d), further suggesting that the enhanced BzOH adsorption could be crucial for the enhanced activity over Au@Pd. Notably, DRIFTS measurements of BzOH on Au$_x$@2.9ML-Pd at different temperatures revealed that the sample with a larger Au core size exhibited a higher desorption temperature of BzOH (Supplementary Fig. 30), in good agreement with the theoretical result (Fig. 1d). According to the aforementioned discussion in the theoretical section, the enhanced BzOH adsorption is mainly attributed to the gradually increased lattice expansion of the Pd overlayer. Therefore, the large Pd lattice expansion on the set of Au$_{6.8}$@$y$-ML-Pd catalysts plays the major role in their high activity. We noted that at a low Pd coverage of ~0.4 ML, the TOFs increased remarkably by approximately 7.7 times from $0.07 \times 10^4$ to $0.54 \times 10^4$ h$^{-1}$ as the Au core size increased from 2.8 to 6.8 nm, while at a high Pd coverage of ~3 MLs, the activity increased by a smaller extent from $1.87 \times 10^4$ to $6.86 \times 10^4$ h$^{-1}$. The relatively larger magnitude of the activity increase on Au$_{NP}$@SubML-Pd than on Au$_{NP}$@2.9ML-Pd might stem from the more pronounced ligand effects at low Pd coverages according to XPS and DRIFTS CO chemisorption measurements (Fig. 3).

Increasing Pd coverage while keeping the Au core at the same size, the activity first increased quickly and reached a maximum activity on Au$_{NP}$@2.9ML-Pd and then dropped quickly as the Pd shell thickness further increased for all four sets of catalysts, manifesting the general

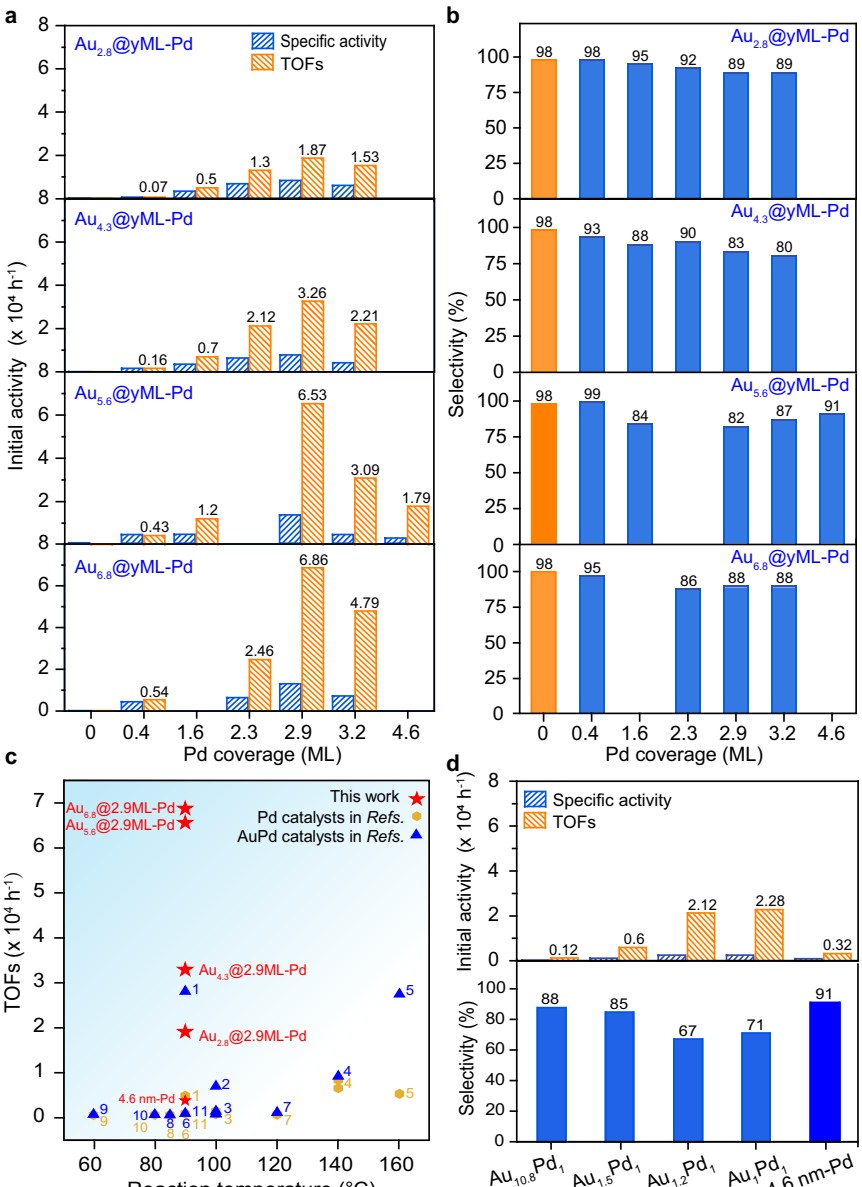

**Fig. 5 | Catalytic performance in the oxidation of benzyl alcohol.** Specific activity and TOFs (**a**) as well as benzaldehyde selectivity (**b**) of the Au$_x$@$y$ML-Pd core-shell catalysts with $x$ nm Au monometallic catalysts. **c** Comparison of the intrinsic activities of Pd and optimized Au$_{6.8}$@2.9ML-Pd catalysts with Pd and AuPd catalysts reported in the literature at different reaction temperatures, where the details of the numbered references are provided in Supplementary Table 6. **d** Specific activity and turnover frequencies (TOFs) as well as benzaldehyde selectivity of the AuPd alloy catalysts and 4.6 nm-Pd monometallic catalysts. Reaction conditions: BzOH, 5 mL; catalyst, 15 mg, except 50 mg for Au$_x$@0.4ML-Pd and $x$ nm-Au catalysts; O$_2$, 15 mL·min$^{-1}$; stirring speed, 1250 rpm; temperature, 90 °C.

remarkable activity regulation by varying the metal shell thickness. Given the obscure Au–Pd interfaces in Au$_{NP}$@$y$ML-Pd core-shell catalysts according to XAFS and CO titration (Fig. 4 and Supplementary Fig. 18), the Au$_{NP}$@2.9ML-Pd catalysts had a Pd shell of ~2–3 MLs. Therefore, the observed activity trend on the Pd shell thickness can be again well rationalized by the calculated adsorption of BzOH, where the strongest adsorption of BzOH on Au(111)@Pd$_{2ML}$ was observed (Fig. 1d). It is worth noting that the TOF achieved on Au$_{6.8}$@2.9ML-Pd after optimizing both the Au core size and Pd shell thickness is approximately 22 times higher than that of the monometallic 4.6 nm-Pd catalyst, the largest degree of activity promotion and is also a record high activity in BzOH oxidation compared to Pd and AuPd catalysts reported in the literature so far (Fig. 5c and more details seen in Supplementary Table 8), even though the reaction temperature here was lower than those used in some references. The above results suggest that the size of the metal core and the thickness of the metal

shell in the bimetallic core-shell structure jointly control the catalytic performance, which has not been reported in the literature to the best of our knowledge.

As a control experiment, we also reduced the four Au$_{6.8}$@$y$ML-Pd ($y$ = 0.4, 2.3, 2.9, and 3.2) core-shell catalysts at 550 °C in 10% H$_2$ in Ar for 1 h to form Au$_{10.8}$Pd$_1$, Au$_{1.5}$Pd$_1$, Au$_{1.2}$Pd$_1$, and Au$_1$Pd$_1$ alloy catalysts, respectively, yet preserving the similar size, as confirmed by the considerable shifts of the diffraction peaks in the XRD spectra[28], TEM and EDS mapping (Supplementary Figs. 31–33). In the reaction of BzOH oxidation, these AuPd alloy catalysts exhibited a large activity decrease compared to their Au$_{6.8}$@$y$ML-Pd core-shell catalyst counterparts, along with slight decreases in benzaldehyde selectivity (Fig. 5c, d and Supplementary Fig. 34). Therein, turning the Au$_{6.8}$@2.9ML-Pd core-shell structure to Au$_{1.2}$Pd$_1$ alloys decreased the TOF considerably from 6.86 × 10$^4$ to 2.12 × 10$^4$ h$^{-1}$. To exclude the possible change in metal-support interactions caused by the high

reduction temperature, which might have influences on the activity, we conducted another control experiment using a lower reduction temperature of 250 °C but with an extended reduction time of 4 h. In fact, a large decline in activity was again observed (Supplementary Fig. 35). Clearly, AuPd alloys are much less active than Au@Pd core-shell NPs, which verifies that a continuous Pd ensemble is necessary for the high activity, in line with the optimized activity achieved on the Au@Pd core-shell catalyst with a Pd shell thickness of ~2–3 MLs.

Regarding the recyclable stability, we found that confining optimized $Au_{6.5}$@2.9ML-Pd core-shell particles in a mesoporous support such as SBA-15 was able to achieve good recyclable stability without any obvious deactivation (Supplementary Figs. 36, 37). Preserving the high activity during the recycling test implies that the structure of Au@Pd core-shell catalysts is not changed under reaction conditions; otherwise, considerable declines in activity would be expected owing to the much lower activity of AuPd alloys than Au@Pd core-shell catalysts (Fig. 5a, c). This is understandable since the reactants of BzOH and $O_2$ both bond more strongly with Pd than with Au. Indeed, STEM measurements of the used catalysts showed that the Au@Pd catalyst retained the particle size as well as the core-shell structure (Supplementary Fig. 38), further confirming the high structural stability under the current reaction conditions.

The conjugated dual size effect found in Au@Pd could be general for core-shell bimetallic catalysis. We further successfully demonstrated it in the Au@Pt-catalyzed selective hydrogenation of para-chloronitrobenzene (p-CNB), another important category of catalytic reactions (Supplementary Figs. 39, 40 and Supplementary Table 9). Therein, we found that by increasing the Au core size from 3.1 to 6.5 nm, but keeping the Pt coverage at 1 ML ($Au_x$@1ML-Pt, $x$ is the average Au core size), the specific activity increased by approximately 7 times from $0.53 \times 10^4$ to $3.65 \times 10^4$ $h^{-1}$. Meanwhile, with variation of Pt shell thickness, but keeping the same Au core size of 5.3 nm, the specific activity also varied considerably.

In conclusion, by combining theoretical calculations with delicate catalyst synthesis and comprehensive structure characterization, we disclosed a conjugated dual size effect in core-shell bimetallic catalysis. In the solvent-free BzOH oxidation reaction, the activities of Au@Pd core-shell catalysts tightly rely on both the Au core size and the Pd shell thickness. By tailoring these two simultaneously, a record high activity was achieved on $Au_{6.8}$@2.9ML-Pd compared to Pd and AuPd catalysts reported in the literature. We showed that a large Au core size along with a thin Pd shell of ~2 MLs is vital for preserving the large lattice expansion of the Pd shell with a high $d$-band center but minimizing the destabilization of the ligand effect from the Au core for high activity. The conjugated dual size effect in core-shell bimetallic catalysis disclosed here can be general and was further successfully demonstrated in Au@Pt-catalyzed selective hydrogenation of p-CNB, another important category of reactions. These findings provide deep insight into the essential roles of the dual particle size effect in bimetallic catalysis, paving the way for the rational design of advanced bimetallic catalysts.

## Methods

### Chemicals and Materials
Tetrachloroauric acid ($HAuCl_4 \cdot 4H_2O$, >99%), palladium acetylacetonate ($Pd(acac)_2$, 99.5%), ammonium hydroxide ($NH_3 \cdot H_2O$, 26–28 wt%), benzyl alcohol ($C_6H_7OH$, 99%) and ethanol (>99.5%) were purchased from Sinopharm Chemical Reagent Co. Ltd. (Shanghai, China). Palladium hexafluoroacetylacetate ($Pd(hfac)_2$, >97%) was purchased from Sigma-Aldrich, and trimethyl(methylcyclo-pentadienyl) platinum (IV) ($MeCpPtMe_3$, >99%) was purchased from Strem Chemicals. All gases, including ultrahigh purity $N_2$ (99.999%), Ar (99.999%), and $O_2$ (99.999%), and mixtures of 10% $O_2$, 10% $H_2$, and 10% CO in Ar (or He), were provided by Nanjing Special Gases. All chemicals were used as received without further purification.

### Synthesis of Pd/SiO₂ catalysts
The spherical $SiO_2$ support was synthesized according to the modified Stöber method[39]. The $Pd/SiO_2$ catalyst was synthesized by the wet-impregnation method. Here, 0.046 g $Pd(acac)_2$ was dissolved into 50 mL acetylacetone. Then, 400 mg spherical $SiO_2$ was added to the solution and mixed under vigorous stirring at 25 °C for 24 h. The obtained solid was dried at 110 °C overnight and further calcined at 500 °C in 10% $O_2$/He for 3 h followed by a reduction step at 250 °C in 10% $H_2$/Ar for 2 h to obtain the $Pd/SiO_2$ catalyst. The loading of Pd was 3.5 wt% according to ICP–AES analysis.

### Synthesis of Au/SiO₂ catalysts
To achieve selective Pd deposition on Au NPs instead of a $SiO_2$ support for the later synthesis of Au@Pd core-shell bimetallic catalysts, a spherical $SiO_2$ support was calcined in 10% $O_2$/Ar at 700 °C for 3 h to remove most of the surface hydroxyl groups to promote selective Pd deposition.

The $Au/SiO_2$ catalysts were prepared using the DP method[17]. The size of the Au NPs was carefully controlled by varying the amount of $HAuCl_4 \cdot 4H_2O$ and adjusting the reduction temperature and pH value by ammonia. These catalysts were denoted as $x$ nm-Au ($x$ is the particle size of Au NPs on average). Typically, for 2.8 nm-Au, 1 mL $HAuCl_4$ aqueous solution (0.0485 M), 1.0 g spherical $SiO_2$, and 150 mL deionized water were co-added into a three-necked bottle and mixed for 30 min under vigorous stirring at 25 °C, and ammonia was used to adjust the pH value between 8 and 9. Then, the system was vigorously stirred for another 12 h. The suspension was then centrifuged and washed with deionized water several times and dried at 80 °C overnight. Finally, the resulting material was calcined at 180 °C in 10% $O_2$/He at a flow rate of 20 mL·$min^{-1}$ for 1 h to obtain the 2.8 nm-Au catalyst. For 4.3 nm-Au, 5.6 nm-Au, and 6.8 nm-Au, the amount of $HAuCl_4$ solution was increased to 3.0 mL, and the ammonia reduction temperatures were 25, 65, and 65 °C with pH values of 8, 10, and 12, respectively. Finally, these materials were all calcined at 250 °C in 10% $O_2$/He at a flow rate of 20 mL·$min^{-1}$ for 1 h to obtain 4.3 nm-Au, 5.6 nm-Au, and 6.8 nm-Au catalysts.

### Synthesis of Au@Pd core-shell catalysts
Synthesis of Au@Pd core-shell catalysts was performed by exclusive deposition of Pd on Au NPs via ALD. Typically, Pd ALD was carried out on the $Au/SiO_2$ samples at 150 °C in a viscous flow reactor (ALD-V401-PRO, ACME (Beijing) Technology) using $Pd(hfac)_2$ and ultra-high purity $H_2$ (99.999%) as precursors[18,19]. Ultrahigh purity $N_2$ (99.999%) was used as a carrier gas at a flow rate of 200 mL/min. The $Pd(hfac)_2$ precursor container was heated to 65 °C to obtain a sufficient vapor pressure. The reaction chamber was heated to 150 °C, and the inlet manifolds were held at 110 °C to avoid precursor condensation. The timing sequence was 120, 200, 60, and 200 s for $Pd(hfac)2$ exposure, N2 purge, H2 exposure, and N2 purge, respectively. Au@Pd core-shell bimetallic catalysts were fabricated by selectively depositing Pd onto $x$ nm-Au catalysts for different cycles. The resulting samples were denoted as $Au_x$@$y$ML-Pd. As a control experiment, Pd ALD was also applied on the bare $SiO_2$ support with different numbers of ALD cycles under the same conditions as described above. All catalysts were calcined in 10% $O_2$/Ar at 150 °C for 0.5 h and then reduced at 150 °C for another 0.5 h in 10% $H_2$/Ar before any characterization and reaction tests.

### Synthesis of AuPd/SiO₂ alloy catalysts
The silica supported $Au_{10.8}Pd_1$, $Au_{1.5}Pd_1$, $Au_{1.2}Pd_1$, and $Au_1Pd_1$ alloy catalysts with different Au/Pd mole ratios were obtained by annealing the $Au_{6.8}$@0.3ML-Pd, $Au_{6.8}$@2.3ML-Pd, $Au_{6.8}$@2.9ML-Pd, and $Au_{6.8}$@3.2ML-Pd core-shell bimetallic catalysts at 550 °C in 10% $H_2$/Ar for 1 h, respectively.

## Synthesis of Au@Pt core-shell catalysts

Au@Pt/SiO$_2$ core-shell bimetallic catalysts were also synthesized by selectively depositing Pt onto $x$ nm-Au samples using Pt ALD[40]. Here, Pt ALD was carried out on the same ALD reactor at 150 °C. The MeCpPtMe$_3$ precursor was heated to 65 °C to achieve a sufficient vapor pressure, and pure O$_2$ was used as the oxidant. The timing sequence was 30, 200, 6, and 200 s for MeCpPtMe$_3$ exposure, N$_2$ purge, O$_2$ exposure, and N$_2$ purge, respectively. The resulting samples were denoted as Au$_x$@$y$ML-Pt.

## Morphology and composition

TEM measurements were performed on a JEM-2100F instrument operated at 200 kV to characterize the morphology of these catalysts. Elemental mapping using EDS was performed on the same equipment. Atomic resolution HAADF-STEM characterization of Au$_{6.8}$@1.1ML-Pd, Au$_{6.8}$@2ML-Pd, and Au$_{6.8}$@2.9ML-Pd bimetallic catalysts was carried out on an aberration-corrected HAADF STEM instrument at 200 kV (JEM-ARM200F, University of Science and Technology of China).

The Pd contents of the resulting Pd catalysts were determined by ICP–AES by dissolving these samples into hot aqua regia. XRD patterns of these samples were collected on a Rigaku/Max-3A X-ray diffractometer with Cu Kα radiation ($\lambda$ = 1.54178 Å), where the operation voltage and current were maintained at 40 kV and 200 mA, respectively. The data were recorded over 2$\theta$ ranges of 20–80°.

## DRIFTS CO chemisorption measurements

DRIFTS CO chemisorption measurements were performed on a Nicolet iS10 spectrometer equipped with a mercury-cadmium-telluride (MCT) detector and a low-temperature reaction cell (Praying Mantis Harrick). After loading a sample into the cell, it was first calcined in 10% O$_2$/Ar at 150 °C for 30 min followed by a reduction in 10% H$_2$/Ar at 150 °C for another 30 min. After cooling the sample to room temperature under a continuous flow of Ar, a background spectrum was collected. Subsequently, the sample was exposed to 10% CO/Ar at a flow rate of 20 mL·min$^{-1}$ for approximately 0.5 h until saturation. Next, the sample was purged with Ar at a flow rate of 20 mL·min$^{-1}$ for another 30 min to remove the gas-phase CO, and then the DRIFT spectrum was collected with 256 scans at a resolution of 4 cm$^{-1}$.

## UV–Vis and XPS measurements

The UV–Vis spectra were measured on a Shimadzu DUV-3700 spectrophotometer. The XPS measurements were taken on a Thermo-VG Scientific Escalab 250 spectrometer equipped with an Al anode (Al Kα = 1486.6 eV). The binding energies were calibrated using the C 1$s$ peak at 284.4 eV as the internal standard. XPS PEAK 4.1 software was used for the deconvolution of XPS spectra. The backgrounds were removed using a Shirley-type integral, and the broadening function was chosen to be a symmetric Voigt function with 80% Lorentzian character. Separated spin-orbit components (Δ = 5.26/17.5/3.7 eV for Pd 3$d$, Au 4$d$, and Au 4$f$ regions, respectively) and the different area ratios for split peaks (3:2 for $d_{5/2}$:$d_{3/2}$ orbitals and 7:5 for $f_{7/2}$:$f_{5/2}$ orbitals) are well fixed for doublet separation. All samples were pretreated with 10% O$_2$/Ar and then 10% H$_2$/Ar at 150 °C for 0.5 h in a tube furnace before UV–Vis and XPS measurements.

## XAFS measurements

In situ XAFS measurements at the Pd $K$-edge (24350 eV) were performed with a Si(311) monochromator at the BL14W1 beamline of the Shanghai Synchrotron Radiation Facility (SSRF), China. The storage ring of SSRF worked at 3.5 GeV in the top-up mode with a maximum current of 210 mA. Ex situ XAFS measurements at the Au $L_3$ edge (11,919 eV) were performed with a Si(111) monochromator at the 1W1B beamline of the Beijing Synchrotron Radiation Facility (BSRF), China. The storage ring of the BSRF worked at 2.5 GeV in the top-up mode

with a maximum current of 250 mA. Considering the contents of Pd and Au, the XAFS spectra of Au$_x$@$y$ML-Pd catalysts at the Pd $K$-edge and Au $L_3$ edge were recorded in fluorescence mode and transmission mode, respectively. All the catalyst powders were first pressed into pellets and then loaded into a homemade quartz reaction cell, where Kapton tape was used as the X-ray window material. This quartz reaction cell can be heated to 500 °C with external heating. A K-type thermocouple, protected by a closed end quartz tube, was located near the sample pellet to measure the sample temperature. After loading into the reaction cell, each sample was first reduced in 10% H$_2$/He at 50 °C for 0.5 h (20 mL·min$^{-1}$). Next, the sample was purged in ultrahigh purity He at 50 °C for 15 min to collect the XAFS spectra.

## Pd dispersion measurements

The Pd dispersions of Pd, AuPd alloy, and Au@Pd core-shell catalysts were determined by CO pulse chemisorption, which was conducted on a Micromeritics AutoChem II chemisorption instrument. After loading a sample, the catalyst was first calcined in 10% O$_2$/He and then reduced in 10% H$_2$/He at 150 °C for 0.5 h. Then, the catalysts were cooled to room temperature in He, and CO pulses were introduced to the catalyst surface using 10% CO/He until saturation. The amount of chemisorbed CO was calibrated by using a thermal conductivity detector. For dispersion calculations, a stoichiometry of CO:Pd of 1:1 was assumed for Au$_x$@SubML-Pd and AuPd alloy catalysts due to the high ratios of linear to bridge-bonded CO in the DRIFTS CO spectra, while a stoichiometry of CO:Pd of 1:2 was applied to 4.6 nm-Pd and Au$_x$@$y$ML-Pd with high Pd coverages according to the literature[41–43].

## Catalyst evaluation

Solvent-free aerobic oxidation of benzyl alcohol using molecular O$_2$ was carried out in a 25 mL three-necked glass flask equipped with a reflux condenser at 90 °C under atmospheric pressure. Then, 5 mL benzyl alcohol and certain amounts of catalyst (50 mg for $x$ nm-Au and Au$_x$@SubML-Pd catalysts, 15 mg for other Pd-based catalysts) were added to the glass flask. Prior to the reaction, the system was first charged with O$_2$ by bubbling ultrahigh purity O$_2$ at a flow rate of 15 mL·min$^{-1}$ for 15 min to remove air. Under the continuous flow of O$_2$, the flask was immersed into a dimethylsilicic oil bath at 90 °C to initiate the reaction. During the reaction, the mixture was vigorously stirred at a rate of 1250 rpm to exclude any mass transfer limitation. Finally, the reaction products were analyzed using a Shimadzu GC-2014 gas chromatograph equipped with a Rtx-1 capillary column and an autoinjector.

The selective hydrogenation of *para*-chloronitrobenzene (*p*-CNB) was carried out in a 100 mL stainless steel batch reactor (Anhui Kemi Machinery Technology Co., Ltd). The amount of catalyst used for the reaction test was normalized to have the same Pt to substrate mole ratio of 1:7800. Typically, *p*-CNB (8 mmol), the catalyst, *m*-xylene (8/3 mmol) and C$_2$H$_5$OH (40 mL) were loaded into the autoclave. Here, *m*-xylene was used as the internal standard. Next, the autoclave was flushed with ultrahigh purity Ar to remove the air and then heated to 65 °C under Ar. When the temperature was stabilized at 65 °C, the autoclave was pressurized with 0.3 MPa hydrogen. Then, the mixture was stirred at a rate of 1500 rpm to start the reaction. During the reaction, the liquid sample mixture was removed through a sampling valve and analyzed by gas chromatography (Shimadzu GC-2014, equipped with an Rtx-1 capillary column and autoinjector).

The initial TOFs of Pd- or Pt-based catalysts were evaluated at conversions below 20% in the kinetic regime unless otherwise noted, according to the following equation (Eq. 1):

$$\text{TOFs} = \frac{\text{Moles of rectant converted}}{\text{Moles of surface metal atoms} \times \text{Reaction time}} \quad (1)$$

Here, the moles of surface metal atoms (Pd or Pt) were determined by CO titration. The specific rates were also calculated using the

following equation with the total amount of Pd or Pt (Eq. 2):

$$\text{Specific rates} = \frac{\text{Moles of rectant converted}}{\text{Moles of total metal atoms} \times \text{Reaction time}} \quad (2)$$

## Computational methods

Periodic, spin-polarized DFT calculations were performed using the Vienna Ab initio Simulation Package (VASP)[44] at the GGA level. The core electrons were described using the projector-augmented wave (PAW) method[45]. Kohn-Sham valence states [Pd($5s4d$), Au($6s5d$), O($2s2p$), C($2s2p$), H($1s$)] were expanded in a plane wave basis set with a 500 eV kinetic energy cut-off. The electronic states were smeared using the Methfessel-Paxton method with a smearing width of 0.1 eV. The optB86b-vdW functional was used to include dispersive forces[46,47]. The equilibrium lattice constants for Au and Pd were calculated to be 4.122 and 3.904 Å, respectively, in close agreement with the corresponding experimental values[48].

For our surface model Au(111)@Pd$_{x\text{ML}}$, which refers to $x$ MLs of Pd supported on Au(111), the bottom two metal layers were fixed at the bulk positions to mimic the core layers, with the rest of the layers being fully relaxed. The adsorption of BzOH was studied on a $(3 \times 3)$ surface unit cell. A $\Gamma$-centered $5 \times 5 \times 1$ Monkhorst-Pack k-point grid was used to sample the surface Brillouin zone[49]. Each slab was separated from its periodic images in the $z$ direction by at least ~10 Å of vacuum. Adsorption was studied only on one side of the slab (adsorbates are fully relaxed), with dipole decoupling applied in the z direction[50].

The classic cuboctahedral shape was taken to model Pd partially substituted Au clusters. One of the outermost (111) layers of the pure Au cluster was substituted with Pd to represent the Au@Pd core-shell structure in an ideal manner. The cluster was separated from its periodic images by ~16 Å of vacuum. The $k$-point grid was sampled at the $\Gamma$-point only for cluster models and gas-phase molecules. Molecules in the gas phase were optimized in a $14 \times 15 \times 16$ Å$^3$ simulation cell. Calculations for both cluster models and gas-phase molecules have dipole corrections applied in all three directions in the unit cell. The criterion for optimization was set to the extent that the maximum residual force was 0.03 eV/Å or less in all relaxed degrees of freedom.

The adsorption energy was calculated as $\Delta E_{ads} = E_{total} - E_{model} - E_{gas}$, where $E_{total}$, $E_{model}$, and $E_{gas}$ refer to the energy of the model with the adsorbate, the energy of the corresponding clean model structure (either a slab model or a cluster model), and the energy of gas-phase adsorbate in a neutral state, respectively. A more negative value indicates stronger adsorption.

## Data availability

The data that support the plots within this paper and other findings of this study are available from the corresponding authors upon reasonable request due to the data are of large amount.

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

## Acknowledgements

This work was supported by the National Key R&D Program of China (2021YFA1502802, 2018YFA0208603, 2017YFA04028-00, 2021YFA1500403), the National Natural Science Foundation of China (22025205, 22221003, 91945302, 12135012), the Fundamental Research Funds for the Central Universities (WK2060000038, 20720220009), Users with Excellence Program of Hefei Science Center CAS (2019HSC-UE016), and K. C. Wong Education (GJTD-2020-15). The authors also gratefully thank the BL14W1 beamline at the Shanghai Synchrotron Radiation Facility (SSRF), the 1W1B beamline of the Beijing Synchrotron Radiation Facility (BSRF) and the Supercomputing Center of University of Science and Technology of China.

## Author contributions

J.L.L. designed the experiments; W.X.L. designed the calculations; X.H.Z. synthesized and characterized the catalysts and performed the catalytic performance evaluation; J.L.L., Z.H.S., and S.Q.W. designed the XAFS measurements; R.J., L.N.C., H.W.W., S.L., X.Y.L., and L.L.W. performed the XAFS measurements; X.H.Z., R.J., Z.H.S., and S.Q.W. analyzed the XAFS data; Y.L. performed the HAADF-STEM measurements; Q.Q.G. and H.C.Y. assisted in the catalyst characterization and catalytic performance evolution; C.W.Z. and C.L.Z. performed the DFT calculations; J.L.L., W.X.L., and X.H.Z. wrote the manuscript, and all the authors contributed to the overall scientific interpretation and edited the manuscript.

## Competing interests

The authors declare no competing interests.
