## [Peer Review File · Nature Communications]

Conjugated dual size effect of core-shell particles synergizes bimetallic catalysisREVIEWER COMMENTS

Reviewer #1 (Remarks to the Author):

The work shows very interesting results regarding the control of catalytic activity with the composition and size of nanoparticles.

Now, the authors claim that the theoretical calculations agree and explain the experimental findings. However, the simulations performed are not representative of the material prepared. The simulations are not of core-shell nanoparticles since only one facet of the Au nanoparticles is replaced by Pd.

The authors need to simulate the core-shell nanoparticles by having a full shell of Pd and a core of Au. Otherwise, one cannot be sure of the agreement between theory and experiment as they are dealing with two different systems.

Reviewer #3 (Remarks to the Author):

This is a very useful paper on a difficult subject of fundamental and applied interest, and I strongly support the publication. I especially like the depth and width of the methods used to investigate this challenging problem.

The authors presented the interesting work of computational prediction of catalytic properties of core-shell gold-palladium nanoparticles towards oxidation of benzyl alcohol. Based on the results authors synthesized the nanoparticles and measured the catalytic activity. Authors provide a deep insight into the properties and morphology of studied materials. The studied reaction is described in every detail necessary to evaluate the advantages of the proposed catalyst. The catalyst designed by the authors exhibits high catalytic activity and selectivity. Also, the stability of a core-shell catalyst is proven to be high.

Only one comment: The comparison between the core-shell and alloy nanoparticles shown in this work may be greatly influenced by the difference in the interaction between the metal and the oxide support, which could be caused by the difference in calcination temperatures. In the synthesis of core-shell nanoparticles presented in this work before ALD of palladium, the material was calcinated at 180 C and 250 C. Alloys, on the other hand, were prepared by heating up to supported core-shell nanoparticles to 550 C. The calcination temperature is known to greatly influence the metal-support interactions, which can further influence the catalytic properties. To compare purely the structural effects of nanoparticles the calcination temperatures should be the same.

Apart from a few minor language mistakes, the article is very well-written.

p. 7. l 151 “contractons” contractions.

p. 23 l. 479 Therein, we found that increasing the Au core size from 3.1 to 6.5 nm, but keeping the Pt coverage at 1 ML (Aux@1ML-Pt, x is the average Au core size), the specific activity increased by approximately 7 times from 0.53×10^4 to 3.65×10^4 h⁻¹.

Consider adding “by” before “increasing”

p. 23 l. 482 Meanwhile, variation of Pt shell thickness, but keeping the same Au core size of 5.3 nm, the specific activity also varied considerably.

Consider adding “with” before “variation”

Commas should be used before "and" at the end of enumerations.

A language touch-up is needed before publication, preferably by an English language editor. This regards mainly the use of the articles, the missing or inappropriate use of 'the' or 'a' and the use of prepositions.

Further review is not needed.

Reviewer #1

Comments:

The work shows very interesting results regarding the control of catalytic activity with the composition and size of nanoparticles.

Response: We greatly appreciate the reviewer for giving us the positive comments on our manuscript.

Now, the authors claim that the theoretical calculations agree and explain the experimental findings. However, the simulations performed are not representative of the material prepared. The simulations are not of core-shell nanoparticles since only one facet of the Au nanoparticles is replaced by Pd. The authors need to simulate the core-shell nanoparticles by having a full shell of Pd and a core of Au. Otherwise, one cannot be sure of the agreement between theory and experiment as they are dealing with two different systems.

Response: We greatly appreciate the reviewer's constructive suggestions.

As suggested, we have performed new calculations to simulate the Au@Pd_{1ML} core-shell nanoparticles by having a full shell of Pd and a core of Au, including Au₁₃@Pd₄₂, Au₅₅@Pd₉₂, Au₁₄₇@Pd₁₆₂ and Au₃₀₉@Pd₂₅₂ (Fig. R1). Calculated adsorption energies and optimized geometries are shown in Fig. R2, where the overall trend behavior is well consistent with our previous calculations with only one Pd facet. As shown in Fig. R2a, on the smallest Au₁₃@Pd₄₂ cluster, the corresponding BzOH adsorption energy of -2.59 eV is the weakest one. For the larger cluster of Au₅₅@Pd₉₂, there is a considerable enhancement in adsorption energy of -2.88 eV. Whereas for further increase of the size to Au₁₄₇@Pd₁₆₂ and Au₃₀₉@Pd₂₅₂, the enhancement becomes small with adsorption energies of -2.96 and -2.90 eV, respectively. Meanwhile, the optimized Pd-Pd distance (in average) increase gradually from 2.72 to 2.75, 2.76, and 2.78 Å for Au₁₃@Pd₄₂, Au₅₅@Pd₉₂, Au₁₄₇@Pd₁₆₂ and Au₃₀₉@Pd₂₅₂, respectively (Fig. R3). The reduction in lattice contraction with the cluster size improves the adsorption, accordingly. In fact, calculated adsorption energies of BzOH on bulk truncated core-shell clusters with larger lattice constants (Fig. R4) are stronger than those on the relaxed structures with smaller lattice constants, in line with above finding. We also calculated the BzOH adsorption energies on the full Pd shell but without presence of the Au core (blue curve in Fig. R2a) and found that calculated adsorption energies become much stronger, illustrating the pronounced destabilization from the Au core, same with previous calculations with only one Pd facet.

In revised manuscript, we have used the full Pd shell model to replace the model with only one Pd facet and updated the corresponding figures and contexts in both main context (pages 5-8) and Supporting Information. All the changes are highlighted in yellow.

Fig. R1. Computational models of cuboctahedral Au@Pd_{1ML} core-shell clusters. (Blue=Pd, gold=Au).

Figure R2. a) Adsorption energies of BzOH on Au@Pd_{1ML} clusters (pink circle) and the counterparts without the presence of the Au core (blue circle). b) Top and side views of the corresponding optimized geometries of BzOH adsorption on Au@Pd_{1ML} NPs (blue=Pd, gold=Au, black=C, white=H, red=O).

Fig. R3. Bulk truncated Au₁₃@Pd₄₂ (a), Au₅₅@Pd₉₂ (b), Au₁₄₇@Pd₁₆₂ (c), and Au₃₀₉@Pd₂₅₂ (d) clusters. Relaxed Au₁₃@Pd₄₂ (e), Au₅₅@Pd₉₂ (f), Au₁₄₇@Pd₁₆₂ (g), and Au₃₀₉@Pd₂₅₂ (h) clusters. The corresponding edge lengths of Pd(111) facet before and after relaxation are indicated. The blue and gold spheres represent Pd and Au atoms, respectively. i) Average Pd-Pd distances in the relaxed clusters.

Figure R3. Adsorption energies of BzOH on Au@Pd core-shell cluster model in bulk truncated and relaxed geometry. The adsorption of BzOH on Au(111)@Pd_{1ML} as reference is indicated.

Reviewer #3

Comments:

This is a very useful paper on a difficult subject of fundamental and applied interest, and I strongly support the publication. I especially like the depth and width of the methods used to investigate this challenging problem.

The authors presented the interesting work of computational prediction of catalytic properties of core-shell gold-palladium nanoparticles towards oxidation of benzyl alcohol. Based on the results authors synthesized the nanoparticles and measured the catalytic activity. Authors provide a deep insight into the properties and morphology of studies materials. The studied reaction is described in every detail necessary to evaluate the advantages of the proposed catalyst. The catalyst designed by the authors exhibits high catalytic activity and selectivity. Also, the stability of a core-shell catalyst is proven to be high.

Response: We greatly appreciate the reviewer for giving us the positive comments on our manuscript.

Only one comment: The comparison between the core-shell and alloy nanoparticles shown in this work may be greatly influenced by the difference in the interaction between the metal and the oxide support, which could be caused by the difference in calcination temperatures. In the synthesis of core-shell nanoparticles presented in this work before ALD of palladium, the material was calcinated at 180 °C and 250 °C. Alloys, on the other hand, were prepared by heating up to supported core-shell nanoparticles to 550 °C. The calcination temperature is known to greatly influence the metal-support interactions, which can further influence the catalytic properties. To compare purely the structural effects of nanoparticles the calcination temperatures should be the same.

Response: We greatly appreciate the reviewer for the insightful comments.

To examine the influence of calcination temperature, we performed another control experiment by decreasing the reduction temperature from 550 to 250 °C, but with an

extended reduction time to 4 h to ensure the formation of AuPd alloy. Here the Au_{6.8}@1.2ML-Pd core-shell catalyst was selected for the extended reduction at 250 °C for 4 h in 10 % H₂, the reduced sample was denoted as Au_{3.5}Pd₁ (Au atoms : Pd atoms = 3.5 : 1). XRD showed that the diffraction peaks were located at 38.2, 44.4 and 64.8° on Au_{6.8}@1.2ML-Pd, assigned to Au (111), Au(200) and Au(220), respectively, confirming the core-shell structure (Fig. R5a). After the extended reduction at 250 °C for 4 h, these peaks shifted considerably to 38.5, 44.8 and 65.1° on Au_{3.5}Pd₁, ambiguously confirming the alloy formation. In the reaction of solvent-free selective oxidation of BzOH, we showed that the activity of Au_{3.5}Pd₁ was only 0.34 × 10⁴ h⁻¹, much lower than that of Au_{6.8}@1.2ML-Pd (0.90 × 10⁴ h⁻¹) (Fig. R5b,c), in an excellent agreement with the trends of those samples reduced at 550 °C. This result clearly verifies that the large activity decline on AuPd alloy catalysts compared with those Au@Pd core-shell catalysts (Fig. 5d) is solely attributed to the change in the catalyst structures instead of metal-support interactions.

In revised manuscript, we have added the above discussion on page 22, the new result was added in Supplementary Information as Supplementary Figure 35. All the changes are highlighted in yellow.

Fig. R5. a. XRD spectra of the Au_{6.8}@1.2ML-Pd core-shell catalyst before and after reducing at 250 °C for 4 h (Au_{3.5}Pd₁). b. Conversion of BzOH as a function of reaction time over the Au_{6.8}@1.2ML-Pd and Au_{3.5}Pd₁ alloy catalysts. c. TOFs of these two catalysts. Reaction conditions: BzOH, 5 mL; catalyst, 15 mg; atmospheric pressure; temperature, 90 °C.

Apart from a few minor language mistakes, the article is very well-written.

p. 7. l 151 “contractons” contractions.

p. 23 l. 479 Therein, we found that increasing the Au core size from 3.1 to 6.5 nm, but keeping the Pt coverage at 1 ML (Aux@1ML-Pt, x is the average Au core size), the specific activity increased by approximately 7 times from 0.53×10^4 to 3.65×10^4 h⁻¹.

Consider adding “by” before “increasing”

p. 23 l. 482 Meanwhile, variation of Pt shell thickness, but keeping the same Au core size of 5.3 nm, the specific activity also varied considerably.

Consider adding “with” before “variation”

Commas should be used before "and" at the end of enumerations.

A language touch-up is needed before publication, preferably by an English language editor. This regards mainly the use of the articles, the missing or inappropriate use of 'the' or 'a' and the use of prepositions.

Further review is not needed.

Response: We greatly appreciate the reviewer for the careful reading, we have changed them accordingly.

REVIEWERS' COMMENTS

Reviewer #1 (Remarks to the Author):

Very nice work. I don't have any further suggestions.

Reviewer #3 (Remarks to the Author):

My concerns regarding the difference in calcination temperature have been addressed. Appropriate additional experiments were conducted. The additional results and discussion on the issue were also included in the revised version of the manuscript. I have no further objections.

The work shows very interesting results of fundamental and applied interest regarding the control of catalytic activity with the composition, structure, and size of nanoparticles. This is a very useful paper and I strongly support the publication.

REVIEWERS' COMMENTS

Reviewer #1 (Remarks to the Author):

Very nice work. I don't have any further suggestions.

Reviewer #3 (Remarks to the Author):

My concerns regarding the difference in calcination temperature have been addressed. Appropriate additional experiments were conducted. The additional results and discussion on the issue were also included in the revised version of the manuscript. I have no further objections. The work shows very interesting results of fundamental and applied interest regarding the control of catalytic activity with the composition, structure, and size of nanoparticles. This is a very useful paper and I strongly support the publication.

Our response: We greatly appreciate the two Reviewers for their previous constructive suggestions, which have significantly improved the quality of this manuscript. We greatly thanks two Reviewers for their positive comments on our work at this time.